# Stress and Adjustment during the COVID-19 Pandemic: A Qualitative Study on the Lived Experience of Canadian Older Adults

**DOI:** 10.3390/ijerph182412922

**Published:** 2021-12-08

**Authors:** Alexandra J. Fiocco, Charlie Gryspeerdt, Giselle Franco

**Affiliations:** Department of Psychology, Ryerson University, Toronto, ON M5B 2K3, Canada; cgryspeerdt@ryerson.ca (C.G.); giselle.franco@ryerson.ca (G.F.)

**Keywords:** COVID-19, pandemic, stress, coping, older adults, resilience

## Abstract

In response to the COVID-19 pandemic, social distancing measures were put into place to flatten the pandemic curve. It was projected that older adults were at increased risk for poor psychological and health outcomes resulting from increased social isolation and loneliness. However, little research has supported this projection among community-dwelling older adults. While a growing body of research has examined the impact of the COVID-19 pandemic on older adults, there is a paucity of qualitative research that captures the lived experience of community-dwelling older adults in Canada. The current study aimed to better understand the lived experience of community-dwelling older adults during the first six months of the pandemic in Ontario, Canada. Semi-structured one-on-one interviews were conducted with independent-living older adults aged 65 years and older. A total of 22 interviews were analyzed using inductive thematic analysis. Following a recursive process, two overarching themes were identified: perceived threat and challenges of the pandemic, and coping with the pandemic. Specifically, participants reflected on the threat of contracting the virus and challenges associated with living arrangements, social isolation, and financial insecurity. Participants shared their coping strategies to maintain health and wellbeing, including behavioral strategies, emotion-focused strategies, and social support. Overall, this research highlights resilience among older adults during the first six months of the pandemic.

## 1. Introduction

On 11 March 2020, the World Health Organization declared the coronavirus disease (COVID-19) a global pandemic. Shortly after, on 17 March 2020, the Ontario government of Canada declared a provincial state of emergency, which resulted in institutional closures and the implementation of physical distancing measures to reduce the spread of the SARS-CoV-2 virus. Later that month, additional measures were implemented by the government to help flatten the pandemic curve, encouraging the public to stay at home (i.e., cocooning) and prohibiting social gatherings of more than five people. With a resulting decline in new COVID-19 positive cases, on 27 April 2020, the government released a plan to reopen the province, while maintaining public health measures including mask wearing, hand washing, physical distancing, staying at home unless absolutely necessary, and the creation of small networks of trusted friends or relatives, otherwise known as “social bubbles”. Although the province continued to slowly reopen during the summer months, rates of infection began to increase once again in August 2020 with the new Delta variant, resulting in a second wave of the COVID-19 pandemic (for an overview of Ontario’s response to COVID-19 between March and July 2020, see [1]).

Amidst the pandemic, concern for the health and wellbeing of older adults has been a dominant focal point, with reports of increased risk for infection and death [2,3,4]. Further, the projected risk of social isolation and loneliness [5] was expected to have a profound effect on the physical and mental health of older adults [6,7,8]. An initial collaborative report by The Irish Longitudinal Study on Aging (TILDA) and ALONE suggested a rise in social isolation and loneliness among older adults within the first four months of the COVID-19 pandemic, beginning on 9 March 2020 [5]. Relatedly, in launching their telephone support helpline to continue serving older adults through their Support and Befriending Service, ALONE received over 26 thousand calls from older adults during this time, 75% of whom lived alone [5]. 

Despite the increased risk for infection and concerns regarding social isolation, a growing body of literature suggests that, on average, older adults have fared relatively well during the pandemic [9,10,11]. In June 2020, the Centers for Disease Control and Prevention surveyed 5412 Americans aged 18 years and older, revealing that adults 65+ years of age reported the lowest level of self-reported anxiety, depression, COVID-19 related trauma, substance use, and suicidal ideation [12]. Additionally, in a survey of 3300 respondents in Quebec, Canada, it was found that older adults 60+ years of age reported the lowest level of psychological distress and the highest perception of emotional wellbeing, compared to other age groups between April and September of 2020 [13]. Finally, in a survey of 1310 adults 18 to 88 years of age, increasing age was associated with lower perceived loneliness. However, negative self-perceptions of aging were associated with greater perceived loneliness and psychological distress during lockdown [14].

Experience of the COVID-19 pandemic among older adults is likely nuanced and modified by various factors, including perceptions of risk and personal resources. According to the Transactional Model of Stress [15], the experience of distress in any given context (e.g., COVID-19 pandemic) is determined by the relationship between a person and the environment that is perceived by the person as taxing or exceeding their resources and endangering their wellbeing. According to the model, a situation may be appraised as a threat or challenge, which interacts with the appraisal of resources to manage potential stressful internal or external demands [15]. Consequently, the objective of the current qualitative study was to gain insight into the lived experience of older adults living in Ontario, Canada, during the initial wave of the COVID-19 pandemic, to capture individual perceptions of the pandemic and resources used to manage its impact. Importantly, the value of qualitative analysis of personal accounts lies within its potential to help better understand the ways in which older adults have managed the pandemic. 

## 2. Materials and Methods

### 2.1. Participants and Procedures

The study was approved by Ryerson University’s Research Ethics Board (#2020-191). Older adults were recruited from the community through online advertisements, including social platforms (i.e., Facebook, Twitter, Kijiji) and an e-newsletter that was shared with newsletter subscribers of the Stress and Healthy Aging Research Lab and community partners (i.e., North York Community House, Bathurst Finch Senior Society). Snowball sampling was also conducted, encouraging older adults to share the study advertisement with their peers. The advertisement read “Older Adults Living in the Community: We want to hear from you about your experience with the COVID-19 pandemic”. Inclusion criteria entailed being 65 years of age or older, living independently in the community, having access to internet or a phone for the interview, and speaking English or French (i.e., the interviewer was bilingual). Participants were recruited between May and October of 2020 and were asked to reflect on their experience with the pandemic since its inception.

Following provision of verbal consent, participants engaged in a 30- to 60-min telephone or virtual interview using Zoom. Before engaging in a semi-structured open-ended interview, participants answered closed-ended questions pertaining to demographic characteristics and health status. Interviews took place from 18 May 2020 to 1 October 2020. A total of 22 interviews were conducted; two of which were conducted over the telephone due to accessibility issues and preference. After the 22nd interview in October, the researchers agreed that no new information was being shared by participants, suggesting data sufficiency. Data sufficiency was then confirmed during the initial coding of the transcripts.

### 2.2. Measures

Questionnaires included a demographic questionnaire to index age, sex, ethnicity, living arrangement, caregiver status, perceived socioeconomic status (i.e., self-reported as low, low to medium, medium, medium to high, high), and religiosity. Participants were asked to list any existing medical and health conditions and to rate their health status on a 5-point Likert scale (0 = poor to 4 = excellent). To assess the presence of depressive symptomatology over the last 2 weeks, participants completed the Patient Health Questionnaire-9 (PhQ9; [16]). Finally, participants were asked to rate their level of worry with respect to catching the SARS-CoV-2 virus on a scale from 1 (no worry) to 10 (high worry). 

Semi-structured one-on-one interviews were audio-recorded and transcribed verbatim. Open-ended questions asked participants to reflect on their experience with the pandemic and its impact on their daily life. The interview guide included the following questions: (1) What does a typical day look like for you? How might this differ from before the pandemic? (2) What are your thoughts on how your personal characteristics or health might impact your experience with the COVID-19 pandemic? (3) How much television/ media do you watch? What are your thoughts about how [the media/news] has been informing the public? (4) In which ways do you feel support, or lack of support, during this pandemic? (5) What are your biggest concerns at this time? (6) How has this pandemic changed your views, or perhaps has resulted in new insights? (7) Any last thoughts you would like to share with us?

### 2.3. Data Analyses

Descriptive analyses were conducted for quantitative data. Qualitative interviews were transcribed and analyzed by the authors using an iterative inductive thematic analysis such that the categories were generated from the data itself through open coding [17]. Following a review of the transcripts, semantic and latent codes were generated to label recurring features of the data. Codes were clustered into themes and subthemes based on similarities and overlap in order to identify significant patterns in the data. The themes were then reviewed during a recursive process, and particularly salient extracts were selected. Transcripts were revisited to ensure that all relevant quotes had been extracted, as well as to help further refine the synthesis of data. 

## 3. Results

### 3.1. Participants

Participants were between 65 and 81 years of age, with an average age of 72.23 years (SD = 4.25), representing the youngest-old and middle-old segment of the aging population. The sample was mostly female (59%) and identified as Caucasian, or of European descent (54%). Two dominant religions represented in the sample were Christianity (27%) and Judaism (32%). A majority of the sample were partnered (*n* = 14, 64%); one participant reported being in a common-law relationship and one participant reported being in a same-sex marriage. Two female participants were widows; six participants (27%) were single, never married. Most participants lived with their spouse, either alone (55%) or with children (9%). Six participants lived alone (27%) and two women were not married, living with a family member. Four participants identified as being a caregiver; one of whom was a secondary caregiver for their mother living in a long-term care residence. With the exception of three participants who were renting their apartment, participants owned their property, either living in a house (45%) or a condominium unit (32%). A majority of the sample (54.5%) identified as having a medium-low to medium-high socio-economic status. 

With respect to health, a majority of participants rated their general health as good to excellent, with an average score of 4.27 (SD = 4.78) on the PHQ-9. Approximately 68% of the sample reported having a medical condition; 23% reported that they were being treated for depression and/or an anxiety disorder; 23% reported presenting with a chronic pain condition including osteoarthritis; 36% reported being treated for hypertension. Additional ailments included thyroid disorder, hypercholesterolemia, diabetes, digestive problems, ulcerative colitis, asthma and other respiratory ailments, and glaucoma. Of note, while seven participants (32%) reported “no medical conditions”, four of these seven participants listed medical conditions, including glaucoma, hypercholesterolemia, and prior prostate cancer (fully recovered). 

Overall, participants did not report a high level of worry with respect to catching the virus (mean score of 3.93, ranging from 1 to 6.5 on a 10-point scale). Please see Table 1 for participants’ characteristics. 

### 3.2. Lived Experience of the COVID-19 Pandemic: Qualitative Findings

Two overarching themes were identified from the 22 transcribed interviews: (1) Threat and Challenges Associated with the Pandemic (Figure 1); (2) Coping with the COVID-19 Pandemic (Figure 2). The first theme was uncovered from a range of perspectives on key experiences that were expressed as being a threat or a challenge stemming from the pandemic by some participants, but not others. The theme of threat and challenges gave rise to seven subthemes, including: the threat of the virus, financial threat, media consumption and fear messaging, the challenge of living arrangement, physical distancing and minimal social interactions, health management and services, and the necessity of using new technology. The second theme exemplified how participants were managing or coping with the pandemic, which translated into behavioral strategies, emotion-focused strategies, and social support. Selected quotes by theme and subtheme are presented in Table 2. 

#### 3.2.1. Theme 1: Threat and Challenges Associated with the Pandemic

##### Subtheme 1: Threat of Contracting the SARS-CoV-2 Virus

All participants reflected on perceptions of threat stemming from the COVID-19 pandemic. Overall, participants did not express extensive worry about personally contracting the coronavirus. Participants largely attributed this low level of worry to their good health and strong immune system, as well as their ability to follow “appropriate precautions” such as mask wearing, hand washing, and physical distancing. However, a few participants shared that they did experience infrequent worrisome thoughts, though nothing that significantly impacted their lives. In particular, one participant who presented with an autoimmune disorder and depression felt that they were more vulnerable, especially since they would not be eligible for the vaccine.

Although participants were not generally worried about their personal risk of catching the virus, some participants shared concern about others’ risk of catching the SARS-CoV-2 virus, especially family members who were immune-deficient or living and working in high-risk environments (e.g., nursing homes). Feelings of frustration and disappointment were expressed towards individuals who were not following the safety recommendations, especially younger generations who were described as selfish, irresponsible, and reckless. Some participants shared that their biggest fear was a continuation of the pandemic with no end in sight. Participants emphasized the need for a vaccine to bring the pandemic to an end, and dreaded subsequent waves and lockdowns.

##### Subtheme 2: Financial Threat

Financial concerns were shared by participants who were not retired, or who were dependent on their non-retired spouse. One widowed woman who self-identified as having a low socioeconomic status shared her greatest concern of affording rent to pay for her apartment without continued income. Another woman feared that her husband’s reduced income would eventually make it challenging to maintain their standard of living. In particular, she felt discouraged that her age minimized her employability despite her skill set and that she was dependent on her husband’s income and pension. Another male participant noted that financial instability was his greatest source of stress during the pandemic. He shared that his income was significantly lower than the previous year and although the government financial relief benefit was greatly appreciated, he had concerns of future financial strain following the expiration of his subsidy at the end of the month. Conversely, participants who were retired shared that they felt financially stable during the pandemic and were grateful for being in their retirement, not reliant on employment.

##### Subtheme 3: Fear Messaging in the Media

A majority of participants reported that the consistently negative and fearful messaging presented in the media caused discomfort, and in some cases, anxiety. While consumption of media increased at the beginning of the pandemic, with some participants commenting on the addictive nature of the news, many participants recognized the media’s negative impact on their wellbeing and chose to decrease their consumption. Many participants expressed feeling discomfort due to chronic fear messaging on the news regarding the enhanced threat of the virus among older adults. Some participants expressed a mismatch between media messaging of increased risk among older adults and their own personal health; many participants felt the messaging was ageist and stigmatizing. Some participants emphasized that they were healthier now, or just as healthy as when they were younger. Some participants also felt the media overemphasized fear messaging and lacked adequate pandemic-related education and health promotion for the public.

##### Subtheme 4: Living Arrangement Challenges

Living arrangements appeared to be an important factor that determined how well older adults were managing the pandemic. While living alone and living in small spaces presented with challenges, those living with others and living in a larger space benefited from their living arrangement. Participants who lived alone, especially those without existing family and “social bubbles”, felt an increased sense of isolation and loneliness due to the inability to connect with others during the lockdown. With institutional closures, such as community centers and shopping malls, older adults who lived alone were stripped of their usual opportunities for social participation. One participant also shared concern with reduced tangible support, such as changing their smoke detector, due to restrictions associated with in-person visitations. In contrast, participants who lived with their spouse or family were less likely to express feelings of loneliness and isolation. Many expressed reliance on their spouse to serve as an emotional buffer and to help balance their mood. Participants living with others emphasized companionship and the importance of having someone else to take care of and spend time with, even if those interactions were of trivial nature.

For those living with others, the living space was an important factor that either positively or negatively impacted their pandemic experience. People living with others in a small space shared experiences of relationship strain due to a lack of personal space. Furthermore, some participants reflected on the difficulty of individual quarantine when living in a small space and the risk of spreading the virus within the household. In contrast, participants living with others in a larger space reflected on the utility of having different areas in the home to spend time away from their loved ones when needed. Having their personal space allowed them to engage in their individual interests during the day, which was especially important for spouses with different interests. Those living in houses also shared how they were able to utilize their outdoor space, such as the front porch or back yard, to safely connect with others while maintaining social distancing.

##### Subtheme 5: Physical Distancing and Minimal Social Interactions

Regardless of living arrangement, a majority of participants reflected on changes in social engagement, many of whom lamented the loss of social connections due to recommended restrictions brought forth by the COVID-19 pandemic. The closure of gyms and community centers impacted participants’ fitness regimes and daily group-based activities. Places of worship were closed, which removed an important community channel for many older adults. Some participants, especially those who self-identified as being “a hugger”, reflected emotively on the challenge of not having physical contact with friends and loved ones. Some participants shared stories of their desperate attempts for physical contact, such as simulating human contact by holding a body pillow.

Participants shared the experience of missing social gatherings with friends, lamenting the loss of social dinners, going to the movies, or going out for coffee with friends. One participant with post-traumatic stress disorder (PTSD) who lived with a family member disclosed that the inability to see their friends made it more challenging to manage their symptoms of anxiety. However, a few participants who lived with their spouse shared they were doing well with minimized social interactions.

Some participants reflected on missing their family, especially their young grandchildren. In some cases, the inability to connect with family was due to border closures and travel restrictions; in other cases, it was due to a mismatch of risk assessment and fear. More specifically, some older adults shared that they were not allowed to see their grandchildren because their children or in-laws assumed they were too fragile to be around the grandchildren.

Some participants had experienced important life events during the pandemic, which were more difficult to endure due to social restrictions. A few participants reflected on the difficulty of physical distancing while mourning the loss of their loved one or feeling helpless while their loved one was in hospital without visitation options. One participant also shared their experience of retiring during the pandemic without a proper celebration to support the transition, resulting in a lack of closure.

##### Subtheme 6: The Challenge of Health Management and Health Services

In reflecting on personal health during the pandemic, a majority of participants expressed no major challenges or changes in their personal health. Few experienced transient sleep problems at the beginning of the pandemic due to initial worries about the virus. However, some older adults with existing health concerns who rely heavily on health services expressed challenges with managing their health and navigating a changing healthcare system during the pandemic. Participants with pre-existing mental health concerns such as anxiety, depression, and PTSD largely attributed worsening of symptoms to a lack of control and social isolation. Furthermore, participants with pre-existing physical health ailments expressed various concerns, such as increased pain and muscle loss due to pandemic-related changes in their exercise regime.

Most participants were satisfied with the level of continued care for their medical conditions. One participant expressed a preference for the changes in their medical care, appreciating the ease of meeting with their doctors on a virtual platform. However, some older adults who relied heavily on healthcare services expressed concern about clinic closures and hesitations about the quality of virtual healthcare.

##### Subtheme 7: Use of Technology: A New Necessity

Given the restrictions on in-person interactions, a majority of participants expressed an increased reliance on and use of technology such as chat groups, Zoom, and Facetime. Many participants reflected on the learning curve associated with technology use and the additional supports needed when setting up Internet connections and online applications. Despite some of the initial resistance and challenges experienced by some participants, older adults recognized the new necessity of technology to remain connected with their social network during the pandemic and also noted unexpected benefits related to the ability to work, worship, and learn online. Still, almost all participants noted that social engagement via technology would never truly replace their need for human connection. Importantly, some participants acknowledged that learning how to use communication tools such as Zoom may be daunting for some older adults who believe that they are “too old to learn” and that the government should provide social assistance to support technology use within this segment of the population.

#### 3.2.2. Theme 2: Coping with the COVID-19 Pandemic

##### Subtheme 1: Behavioral Strategies

To minimize the distress associated with media consumption, a number of participants shared that they pointedly reduced their consumption of the news. Those who wanted to stay informed reduced consumption to once or twice a day; one participant chose to peruse newspaper headlines as a way to stay informed, but not become overwhelmed.

A majority of the sample reflected on the responsibility of following COVID-19 health and safety regulations. Wearing a mask, hand washing, and physical distancing were seen as important behaviors to aid in curbing the spread of the coronavirus. Following the recommended practices was seen as an important civic responsibility, and contributed to a sense of personal safety. Persons with compromised immune function, or those living with an immune-compromised spouse, highlighted the importance of wearing a mask despite the discomfort. One participant with asthma and partial facial paralysis shared their experience of actively searching for a mask that was relatively comfortable, not only to protect their own wellbeing, but to protect others as well. This level of social consciousness was expressed by a number of participants in the sample. Despite experiencing discomfort with wearing a mask, one participant with asthma shared that wearing a mask sent a message of social responsibility to others in their community.

Behavioral strategies such as maintaining a daily schedule were an important coping strategy for many older adults. Participants also spoke to the importance of staying busy and finding new hobbies, such as gardening, baking, quilting, reading, and painting. Some participants shared the importance of pivoting, finding new opportunities to replace activities in which they were no longer able to engage. For example, one participant shared their ability to pivot from going to the gym to creating a new art portfolio. Some participants viewed these activities as contributing to their wellbeing while others simply viewed them as a means of distraction. A few participants who did not have varied interests or hobbies before the pandemic seemed to experience a greater challenge with the pandemic and finding purpose in their daily life. One participant without directive interests or hobbies likened the pandemic to Groundhog Day, with an overconsumption of television each day with their husband.

Keeping physically active was another common behavioral method of coping with the pandemic and maintaining wellness. Some older adults shared that their new “free time” allowed them to be more physically active than before the pandemic. In addition, simply being outdoors was perceived as an important strategy for maintaining wellness, especially for those living in smaller spaces.

##### Subtheme 2: Emotion-Focused Strategies

All participants shared the various emotion-focused strategies that were used to help manage living with the pandemic. Gratitude for small pleasures, acceptance of the current situation, and optimism for the future was commonly shared by older adults. Furthermore, perspective-taking was notable in persons who had previously experienced hardship, trauma, or loss. For example, one participant who was a transplant patient noted that we have a choice to enjoy life, regardless of the current circumstances. Another participant shared that the pandemic was not as stressful as the hardship they experienced as a new immigrant to Canada. Taking a more general perspective, one participant reflected on what humanity has already suffered and survived, including war, the plague, and the great depression.

Meaning-making and positive re-appraisal were important strategies shared, allowing older adults to find the silver lining in their pandemic experience. Participants shared how the pandemic created new avenues for skill development, self-reflection and creativity, and how it allowed them to strengthen their relationship with family members. For some, the pandemic brought a sense of respect for the community as they watched people “step up to the plate” to adjust to new circumstances. Still, a few participants reflected on the need for greater social justice with respect to racism and the treatment of older adults living in residential care and nursing homes. One participant shared that their duties had become “more clear” as an anti-racism social activist.

##### Subtheme 3: Social Support

A majority of participants reflected on the importance of social support during the pandemic. Receiving support from family and friends, including both emotional and tangible support, was greatly valued by participants. As noted above, older adults living with their spouse or family were grateful for the consistent companionship during lockdown. One participant reflected on the support received by their children who lived overseas, which included special “care packages” with traditional Korean snacks and cards from the children and grandchildren.

Half of the sample reflected on their social bubble and the valuable support from neighbors and friends. Some participants shared their attempts to maintain safe social gatherings with friends outside, on the porch in their back yard, or on their front lawn to adhere to government regulations. Some participants commented on how neighbors have helped through porch drop-offs, running errands, and sharing items such as books. Further, participants detailed how friends have supported them through regular calls and virtual meetings. Some participants emphasized the importance of connecting with friends as a way of not feeling alone in “pandemic worry”. Importantly, interviews also revealed that older adults living alone might necessitate a greater need and quantity of social support than is regularly available.

Participants not only reflected on how they were being supported, but also shared the importance of supporting others and the ways in which they were supporting friends, family, and their communities during the pandemic. A few participants expressed how they have become more active in their communities by reaching out to others through regular phone calls, and the sense of purpose this has brought to their own lives. Yet, other participants reflected on the challenges of supporting others during the pandemic. For example, one participant expressed guilt over not being able to assist her essential worker colleagues during the pandemic, and another participant expressed difficulties in supporting her mother who was living in a nursing home.

## 4. Discussion

Within the first few months of the pandemic, communities across the world faced a momentous disruption to their livelihood. Although initial reports suggested that older adults were at increased risk of psychological distress stemming from social isolation and loneliness [5], emerging evidence suggests that older adults have displayed greater emotional resilience during this pandemic, relative to younger age groups [5,13]. The present study provides a qualitative perspective of the lived experience of community-dwelling older adults within the first six months of the COVID-19 pandemic. Most notably, the current qualitative results support existing quantitative and qualitative findings of resilience among older adults during the pandemic.

Despite targeted fear messaging on the news and social media, suggesting an increased risk of viral contraction among older adults, participants did not express heightened concern regarding their personal risk of catching the virus due to their age. However, the news media portrayal of COVID-19 as an age-related disease created anxiety in some participants. While media consumption initially increased among many older adults in order to remain informed of the evolving coronavirus and the recommended safety practices, participants also noticed the negative impact that media consumption was having on their emotional wellbeing. Recognizing the need to regulate media consumption, those who chose to decrease their media intake shared that they experienced an improvement in their psychological wellbeing. These findings are aligned with previous research pertaining to COVID-19 media consumption. In a study by Losada-Baltar et al. [14], greater time devoted to COVID-19 information was associated with greater psychological distress during lockdown. Furthermore, data from the American Trends Panel, which sampled adults aged 18 years and older, revealed that greater COVID-19 media consumption was associated with greater psychological distress, which was partially explained by increased perceptions of COVID-19 threat [18]. Together, results highlight the importance of recommending guidelines surrounding media consumption. More specifically, qualitative findings underscore the importance of being an active participant in media consumption, which includes bringing awareness to how specific media are influencing personal wellbeing, and the ability to disengage from counterproductive behaviors. Furthermore, targeted fear messaging may prompt ageist sentiments and behaviors among younger individuals, as some older adults expressed a mismatch between their personal perceptions of wellbeing and their children’s perceptions of risk due to age. This led to frustrations over feeling constrained in seeing family members and running errands, infringing on their desire to maintain independence.

Ageist attitudes have undeniably been exposed during the pandemic. However, the dialogue of systemic ageism during the pandemic has largely focused on nursing homes and older adults living in residential care, without considering the experiences of ageism among community-dwelling older adults. In a qualitative study of 19 older adults’ lived experience during the first COVID-19 lockdown in southern Switzerland, some older adults felt stigmatized in being portrayed as part of the vulnerable, and in some cases undesirable, population [19]. This sense of stigma was expressed in the current sample, with one participant not wanting to leave their house out of fear of being judged. This research highlights the negative ramifications of portraying older adults, a heterogeneous group, as an at-risk population, which promotes social exclusion, stigmatization, and victimization.

General low concern for contracting the SARS-CoV-2 virus among participants was commonly attributed to a sense of one’s personal health or to the engagement in prescribed cautionary behaviors. Older adults shared that they felt healthy, with a strong immune system; some expressed that they were just as healthy now as when they were younger. Those with a compromised immune system felt confident in the prescribed safety measures, emphasizing the importance of physical distancing, hand washing, and the wearing of masks. Due to the prevalence of contentious discourse around mask wearing, many participants conveyed their strong belief in the importance of wearing a mask, not just for personal safety, but for the safety of others. Indeed, following the prescribed safety measures was viewed as a civic duty. Many participants expressed disappointment and anger towards individuals who did not follow the prescribed safety measures, with a common focus on younger generations. This tension with noncompliant citizens and youth aligns with the aforementioned qualitative study in southern Switzerland, which reported anger towards those who did not respect COVID-19 prevention measures as well as the institutions that were not enforcing the rules [19]. This cohort of older adults also expressed anger towards the youth, perceiving a sense of ageism from this generation and a disregard for the lives of vulnerable persons [19]. In the current sample, greater concern for others’ wellbeing was emphasized among older adults, which is aligned with the notion that aging is associated with greater altruism [20]. Considering that altruistic attitudes predict positive effects in later adulthood [21], caring for the personal safety of others and engaging in prosocial behaviors (discussed below) during the pandemic may reflect a characteristic of resiliency among older adults.

Due to institution closures, participants who were still working expressed financial concerns. Older adults who were dependent on employment income were concerned about their ability to pay the rent and to sustain their current standard of living. Although these concerns were shared among a minority of the participants within the current sample, it is important to recognize that some older adults continue to work in later adulthood out of necessity [22]. Further, these stories reflect the economic hardship experienced by many individuals, with exacerbated concerns surrounding housing insecurity due to the pandemic, which may negatively impact health outcomes [23]. Conversely, those who were retired commonly expressed gratitude for being financially secure in their retirement. Many older adults acknowledged that the common pandemic stressors experienced by Canadians, including loss of employment and homeschooling young children, were not a concern for older adults, which may partially account for quantitative findings of increased perceptions of stress among younger adults relative to older adults [13].

Participants shared the challenges associated with social distancing measures and institution closures. Within the current sample, stories were very similar with respect to loss of social connection and daily routine due to the closure of community centers, gyms, and places of worship. Similar to a qualitative study in older adults living in the UK [24], participants lamented the loss of the “old normal”, and longed for activities that they enjoyed, including outings with friends and travel. Stories of loss and bereavement during times of social confinement highlighted the importance of social connectedness during important life transitions—a testament to the value of community and family support. Overall, participants expressed a sense of being supported during the pandemic. Participants living with a spouse expressed gratitude for having someone to spend time with; however, living space was highlighted as an important factor for managing the pandemic day to day, with larger spaces affording the benefit of having moments of solitude. Research to date has largely focused on the effect of the pandemic on indices of loneliness, without considering the need for solitude. Although the need for solitude may differ by age, personality, and circumstance, solitude has been found to support the freedom to engage in personal interests and to foster creativity and spirituality [25].

Many older adults shared a deep sense of community with their neighbors, reflecting on times spent socializing at a distance, helping one another with running errands, or sharing resources. Further, engaging in social outreach activities in their communities provided older adults with a sense of purpose during the pandemic. A sense of purpose in later adulthood has been shown to support healthy lifestyle behaviors [26] and is a cornerstone of wellness in late life [27]. Altogether, stories of personal lived experiences underscore the importance of social capital, including neighborhood cohesion and norms of reciprocity, which may facilitate a sense of purpose and minimize the potential isolating effect of the pandemic. Indeed, engagement in prosocial behaviors as means of communal coping may also foster wellness within the community [28].

While the worry of contracting the virus was minimal within the sample, some participants with existing psychological health ailments (e.g., obsessive-compulsive disorder, post-traumatic stress disorder, depression) expressed that their symptoms were initially exacerbated by the pandemic. Perceived exacerbated symptoms within the initial months of the pandemic were also reported in qualitative interviews among older adults with major depressive disorder in the United States who were enrolled in a clinical trial prior to the pandemic [29]. However, qualitative reports also highlighted resilience in response to the pandemic, and quantitative scores on indices of depression and anxiety did not significantly differ from pre-pandemic scores [29]. Accordingly, ensuring that proper health support systems for persons with existing health ailments are essential. Importantly, none of the participants shared the onset of new psychological or physical diagnoses resulting from the pandemic, which is aligned with the aforementioned quantitative report by Dionne et al. [13]. Rather, participants shared various strategies employed that helped them manage day-to-day pandemic living, an indication of resilience and ingenuity. With regards to access to healthcare, a notable portion of the sample discussed their experience with adapting to changes in access to medical services. Many participants were thankful to be in relatively good health and, therefore, able to adapt to delays in visits with their medical specialists. Of note, one participant who lived alone and was particularly reliant on their family doctor was very disappointed with the closure of their clinic, which was located in a lower-income neighborhood. Although this was one older adults’ experience, it highlights existing inequitable access to care between high and low-income Canadians [30].

A myriad of coping strategies recounted across interviews highlights the importance of adaptive behavioral and emotional coping strategies to mitigate the potential negative impact of pandemic stress on wellbeing. Behavioral strategies were important for the management of institution closures. A majority of older adults, including those with mobility issues, spoke about keeping physically active during the pandemic. Such behavior is supported by a large body of research that highlights the importance of physical activity for physical and emotional wellbeing among older adults, including resistance to negative aging stereotypes [31,32]. As noted above, a number of participants expressed low worry about contracting the virus, which was frequently attributed to their sense of good physical health and hardiness. Accordingly, the ability to remain physically active during the pandemic may support a sense of physical and emotional wellness, even among persons with existing health ailments. This is aligned with a qualitative study among older adults in France, which highlighted the importance of finding ways to support older adults in remaining physically active at home through the development of national policies [33].

While the ability to pivot daily activities was shared by a majority of the sample, one participant shared difficulty finding new ways to shift their activities of daily living. They found that they had “no purpose” and were unable to draw on existing interests or hobbies. According to self-determination theory, the ability to engage in internalized goal-directed behaviors is largely dependent on the need for autonomy, competence, and relatedness [34]. Consequently, persons who lack a sense of autonomy and choice during the pandemic may find it more difficult to rely on internal motivations and to pivot their daily activities to support their wellbeing. This may be especially salient for individuals who rely on external sources of pleasure and gratification [34].

A range of adaptive emotion-focused coping strategies was shared across interviews, all of which support wellbeing. A prominent emotion-focused strategy that was used among participants was perspective-taking, which afforded a sense of strength in overcoming adversity. Examples of gratitude, meaning-making, and positive reappraisal were commonly noted in the qualitative analysis, all of which are interconnected and foster positive emotions. These findings align with a qualitative study in a sample of Midwestern American older adults aged 70 to 97, which reported the use of adaptive emotion-focused coping strategies, including seeking social support and having a positive mindset, to foster resilience during the early weeks of the pandemic [35]. According to Fredrickson’s broaden-and-build model, the cultivation of positive emotions such as gratitude is a cognitive-emotional process that broadens the individual’s thought–action repertoires, which over time fosters the building of emotional, social, and physical resources, which in turn enhance wellbeing [36]. Finally, participants expressed the ability to let go of the desire to change what was outside of their reach, focusing instead on the smaller changes and adaptations that they had the power to make within their daily lives. Shifting focus away from what is outside of one’s control toward that which is controllable is important for wellbeing as it fosters healthy lifestyle behaviors, psychological wellbeing, and physical health [37].

Silver linings around the pandemic were shared by many, including the need and benefits of learning a new tool, namely Zoom. Although resistance to the use of new technology varied, many were grateful that they had a new means of communicating with loved ones and a way to continue engaging in social and learning activities. Previous research suggests that internet use among older adults supports wellbeing and is associated with reduced loneliness and greater life satisfaction and psychological wellbeing [38]. However, resistance to learning new technology was present among some older adults due to novelty and the lack of confidence to use new technology. Similar qualitative findings were found in a sample of older adults living in Northern Italy, with varied uptakes in technology to minimize the effects of social isolation during the first wave of the pandemic [39]. In this study, some participants spoke to accessibility issues due to impairments in vision and hearing [39]. The stories shared by older adults point to the importance of telecommunication support, including accessibility and skill development. While characteristics of aging cohorts continue to change, it is important to address aging stereotypes within the community, including the well-known “I’m too old to learn” stereotype, which hinders healthy aging [40,41]. Continuing education and social programming is essential to breaking down barriers that are created by aging stereotypes, and to foster skill development in later adulthood. Finally, the co-development of technology by older adults may enhance accessibility issues for older adults with age-related impairments in vision and hearing.

Despite the valuable contribution of this report to the growing body of literature, it is important to reflect on study limitations. First, although data sufficiency was met following interviews with 22 participants, all participants resided in the province of Ontario, Canada. While participants lived in both urban and rural environments, the experience of older adults in Ontario may not be representative of other geographical locations. Second, although the authors intentionally worked towards recruiting an ethnically diverse sample, a majority of the sample was Caucasian, with a select few who identified as being part of a minority group. As social determinants influence health and wellbeing [42], it is important to consider how the overrepresentation of some ethnic groups in essential workplaces, as well as financial insecurity and multigenerational living may differentially impact the experiences of older adults during the pandemic. Similarly, the age of participants ranged from 65 to 81 years, therefore, not representing adults within the oldest-old (i.e., 85+ years) segment of the population. Older adults 85+ years of age may present with different experiences as it relates to health, leisure, and the use of new technology during the pandemic. However, given the qualitative nature of the study design, this analysis does not attempt to generalize findings to the greater population. The value of qualitative analysis is in the ability to provide a descriptive snapshot of the experience of a given sample. A final limitation pertains to the timeline of the interviews, all of which were conducted within the first six months of the pandemic, spanning from April to October 2020. Although this allowed for a range of experiences, including the initial lockdown, the gradual reopening of the province, and resurgence of COVID-19 cases with a second wave of the pandemic, it is unclear whether the activities and coping strategies employed during the first six months of the pandemic continued into the winter months and during the extended waves of the pandemic.

## 5. Conclusions

Overall, the lived experiences shared by older adults in the current study are largely reflective of recent quantitative studies suggesting that older adults have been relatively resilient during this pandemic. They did not feel that they were at increased risks of contracting COVID-19; they did not report the onset of new health-related concerns within the first six months of the pandemic; they did not feel a need to be treated differently due to their age. What has surfaced during the pandemic is a need to break down age-related stereotypes that compromise the livelihood and wellbeing of aging Canadians; the need to engage in careful reflection on media intake and its impact on emotional wellbeing; the need to develop and foster prosocial behaviors and communal coping to support wellness within the community; the need to establish accessible health support systems for persons with existing mental health ailments; and the need to create social programming that increases access to telecommunications at any age.

## Figures and Tables

**Figure 1 ijerph-18-12922-f001:**
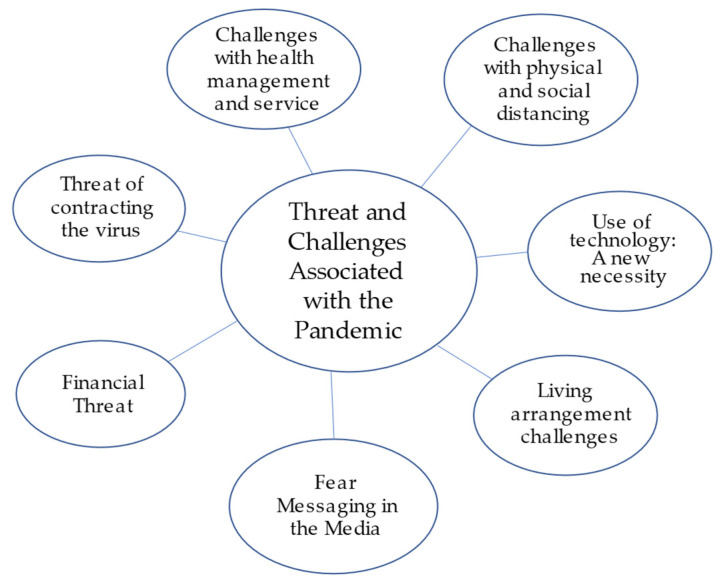
Theme 1 and subthemes.

**Figure 2 ijerph-18-12922-f002:**
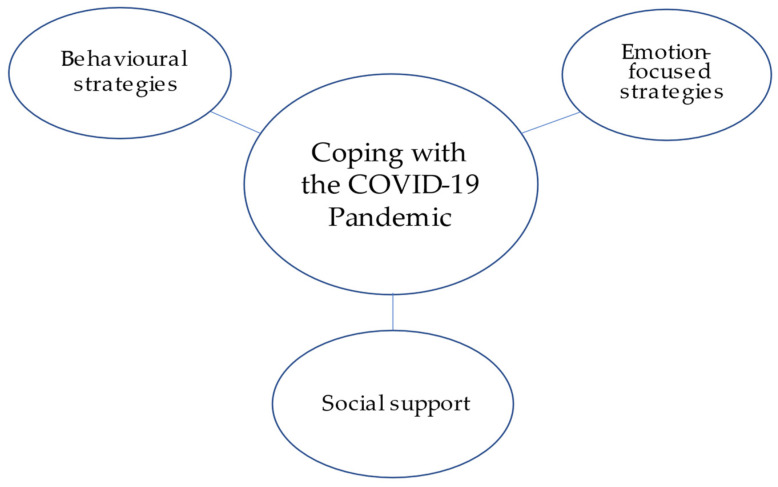
Theme 2 and subthemes.

**Table 1 ijerph-18-12922-t001:** Participants characteristics (*n* = 22).

Descriptive Variable	Mean (SD, Range) or *n* (%)
Age (years)	72 (4.78, min 65–max 81)
Sex (% female)	13 (59)
Ethnicity	
Caucasian/European	18 (82)
Black	1 (5)
Asian	2 (9)
Middle Eastern	1 (5)
Relationship Status	
Married	13 (59)
Common Law	1 (4)
Single	6 (27)
Widow	2 (9)
Living Arrangement	
Alone	6 (27)
Spouse	12 (55)
Spouse and Children	2 (9)
Other family	2 (9)
Place of Residence	
House/Townhouse Owned	10 (45)
Condominium (Owned unit)	7 (32)
Apartment (Rental unit)	3 (14)
Group Housing	2 (9)
Caregiver Status	
Not Caregiver	18 (81)
Primary Caregiver	2 (9)
Secondary Caregiver	2 (9)
Socioeconomic Status	
Low	1 (5)
Medium Low	6 (27)
Medium	7 (32)
Medium High	6 (27)
High	1 (5)
Prefer not to say	1 (5)
Religion	
Christianity	6 (27)
Buddhism	1 (5)
Judaism	7 (32)
Spiritual	5 (23)
Not religious	3 (14)
General Health Status	
Poor	1 (5)
Fair	2 (9)
Good	7 (32)
Very good	8 (36)
Excellent	4 (18)
Existing Medical Conditions	
No	7 (32)
Yes	15 (68)
PHQ-9 score	4.27 (4.78, 0–17)
Worry for SARS-CoV-2 contraction	3.93 (1.52, 1–6.5)

Note: SD = Standard deviation; PHQ = Patient Health Questionnaire.

**Table 2 ijerph-18-12922-t002:** A select set of participant quotes by theme and subtheme.

Theme 1: Threat and Challenges Associated with the Pandemic	
codes	Subtheme 1: Threat of Contracting the SARS-CoV-2 Virus
Low worry (good health)	I’m really not that concerned if I catch it, I catch it, I’m hopeful that I would be able to pull through. I have a fairly good immune system when I get a cold. I do have asthma, and when I get a cold, it does flare it up, but I’ve always managed to control it so it’s not severe enough that it really would worry me. (#14, 70 y/o Irish Jewish female, lives with spouse in condominium)
Low worry (good health)	Compared to others of my age, I feel that I will not be affected by the virus. I won’t get it. (#1, 75 y/o Filipino male, lives with spouse and children in townhouse)
Low worry (good health)	Well even though technically I’m a diabetic, because I’m now exercising much more frequently, I’m now feeling fitter than I have ever done for 20 years. You know, I’m feeling fitter than I was in the mid-50s, so it’s just remarkable…even though technically I should be at risk, but the reality is that I’m feeling quite fit. (#13, 73 y/o Jewish male, lives with spouse in a condo)
Low worry (taking precautions)	I’m not worried about getting the virus. If I take appropriate precaution I think it’s practically zero…it’s unlikely that I’m going to get the virus unless someone I come in contact with has it and I break the rules. (#12, 70 y/o Caucasian male, lives with spouse in a house)
Low worry (taking precautions)	Well, I’m taking precautions, all the precautions like wearing face masks and washing my hands. We’re changing the hand towels much more frequently and we use our own separate hand towels. We’re taking these types of precautions to try to preserve ourselves. (#13, 73 y/o Jewish male, lives with spouse in a condo)
Low worry (taking precautions)	I think [the recommended safety practices] are a precaution that’s necessary. I think we have something we can do to make us feel more secure. I put coffee filters in my in my mask too and I change those to keep them affective, to keep the mask effective. (#7, 81 y/o Caucasian male, living with same-sex spouse in an apartment)
Infrequent worry	I try and take all the necessary precautions… but at the same time I do hear people get the COVID and they cannot figure out how they get it, so that can get me a little uneasy sometimes but it’s not something I worry about, but in the back of my head I’m always saying well OK I’m doing everything right and everything I’m supposed to do, I hope that’s good enough for me not to get COVID. (#18, 78 y/o Black female, lives alone in a house)
Infrequent worry	Because I have an autoimmune disease and I have a disability and stuff, I’m more vulnerable than if I was 25 years old. I have to be more cautious and you know make sure that I’m following the social isolating, and washing my hands a lot, wearing the masks because, with the illness that I have I’m never going to be able to have the vaccine, because I got my illness from having a vaccine from the flu shot, so I have to really protect myself from getting COVID. (#8, 71 y/o Caucasian female, lives with family in group housing)
Infrequent worry	Say I’m having a bit of nausea because of my stomach surgery, sometimes I think ‘Oh my God am I getting COVID’ and then when I wake up I say ‘no no no it’s unlikely’. (#10, 72 y/o Celtic Jewish female, lives with family in group housing)
Concern for others	My daughter’s job has been affected you know like 100%. I mean it’s amazing the changes she’s had to go through and right from the very beginning, she is in a very vulnerable job which has been a worry you know, she’s more exposed from frontline workers. (#13, 73 y/o Jewish male, lives with spouse in a condominium)
Concern for others	I’m glad that I found that [cloth mask] because I feel protected and I’m protecting other people from me too because, my mouth is paralyzed and a lot of times spit will come out of my mouth unintentionally, so I’m protecting other people when I’m out too… I have a good social conscience, I don’t want to make anybody else sick and I don’t want to get sick. (#8, 71 y/o Caucasian female, lives with family in group housing)
Concern for others	We’re also more careful because his [husbad] immune system is shot for the next two years and then me too, like being able to go see my mom. She’s 98 right, so not being able to go to the nursing home and having to worry about her… if I can go there I gotta be more careful as to where I go or who I see because I could be bringing it [the virus] to them. (#4, 69 y/o Caucasian female, lives with spouse in a house)
Frustration with others	From what I see everybody’s following the rules, where it’s the young people who don’t seem to get it… They don’t seem to take it as seriously. I’ve noticed other people’s kids, in their 20s as well, jumping into cars with their friends and driving off, and I’m thinking ‘oh my god’. (#22, 65 y/o Caucasian female, lives with spouse in a house)
Frustration with others	I’m a little bit annoyed that some people think that they shouldn’t do it [i.e., follow safety recommendtations] because of their own individual rights and, you know, it’s just silly. I mean it’s something in a case like this, I think individual rights get suspended and you have to do the right thing for everybody else. (#3, 67 y/o Caucasian female, lives alone in a house)
Frustration with others	When it’s clear that distancing, washing hands, wearing a mask and not gathering in large groups works, why people on God’s green earth think that they can be doing this appalls me. (#16, 70 y/o Jewish male, lives with common law partner in a condominium)
Frustration with others	What pisses me off is when I go grocery shopping, there are a lot of shoppers who don’t wear their mask, or if they do it under their chin or below their nostrils. That really pisses me off. (#11, 74 y/o Jewish male, living with spouse in a condominium)
Fear for continued pandemic	I fear that the virus is going to continue… that we’re not going to find a way to deal with it, like we’re not going to find a vaccine early enough and the that this will just go on. (#15, 65 y/o Ukrainian female, living alone in a house)
Fear for continued pandemic	I worry we will get a second wave, and it won’t be in the fall, it will come this summer. And then we’ll get a third wave. I guess that it will be never ending. (#22, 65 y/o Caucasian female, lives with spouse and children in house)
Fear for continued pandemic	I worry that things will not be as they were, that we won’t go back, where there’s going to be a total new normal in terms of traveling and being able to see people and be comfortable with people. I think that’s my biggest concern, that just things will not be as they were. (#6, 71 y/o Caucasian female, lives alone in a house)
codes	Subtheme 2: Financial Threat
Housing	My biggest concern... that financially I can stay in my apartment, and I can afford it. (#2, 74 y/o Caucasian female, widow lives alone in a condominium)
Standard of living	My biggest concern is probably financial because I worry that my husband still works as a [occupation] and he needs to go to see clients and the clients aren’t as busy. I am 70, they wouldn’t employ me any longer and it’s not, I mean, I worked as a [profession], I haven’t lost that ability, but they just don’t see it… in that sense, that bit upsets me and so I have to rely on my husband’s income and on our pensions, and that it’s tough. It’s tough to live in a fixed income these days. (#14, 70 y/o, Irish Jewish female, lives with spouse in a condominium)
Financial stability	I’m a sole practitioner and at least for April and the beginning of May my work suffered terribly it was very challenging in terms of income and the word I use is discombobulating in terms of working a little bit at home, sometimes at home, sometimes in the office, and that was really challenging. (#16, 70 y/o Jewish male, lives with common law partner in a condominium)
Financial stability	I have a financial concern, because I’m semi-retired…so I don’t get pension income… and I’m concerned since my employment income is significantly lower than it was at this point last year. And I have taken advantage of the government’s subsidies…but you know that’s finishing this month. (#13, 73 y/o Jewish male, lives with spouse in a condominium)
codes	Subtheme 3: Fear Messaging in the Media
Fear messaging	I as a CNN junkie. I’m now not doing that because I don’t think that’s healthy either… I know fear of catching the virus is fueled by media coverage because now that I don’t listen to it, like very little, I’m much happier. Yeah. I think at first it was just like it was like an addiction. (#5, 74 y/o Caucasian female, lives with spouse in a house)
Fear messaging	Every time I watched the news I would just feel my heart sink a bit … I wanted to stay informed as best I could, but I found that it just got to be too much so I limited it to twice a day probably. (#6, 71 y/o Caucasian female, lives alone in a house)
Fear messaging	I think the media has a lot to do with people’s fears… I think sometimes that the media plays it up. So sometimes it’s good not to watch the news … we’re trying very hard not to get too excited about this whole thing. (#9, 73 y/o Caucasian male, living with spouse in a house)
Ageist fear messaging	When it all first came out... a lot of what we heard was that it was going to affect ‘us’ more than anyone else, and that played on my brain a little bit…. all you heard was ‘be careful of the’, you know, ‘it’s the seniors’. (#5, 74 y/o Caucasian female, lives with spouse in house)
Ageist fear messaging	All we heard was it definitely is going to impact people over the age of 65 or 70, and stay home. Basically, we don’t have enough ventilators for you folks—you need to flatten the curve because you’re the ones that are really going to put a stress on the system. I had a social obligation. Like, even if I did poke my head out the door it’s like ‘Oh my gosh people are going to know I’m over 65′ and they’re going to probably look down on me because I’m not in my house. So, for a while I was afraid to go out for a walk even. (#6, 71 y/o Caucasian female, lives alone in a house)
Ageist fear messaging	Everybody says that if you’re 60 or older you are more in danger. I don’t believe in living my life in fear. I mean, I know that I’m 70, but in reality most of the time, other than with my arthritis, I don’t feel any different than I did when I was bringing my kids up… And my age doesn’t, as far as I’m concerned… it’s only a number. (#14, 70 y/o, Irish Jewish female, lives with spouse in a condominium)
Media lacking	Because [friend] is on the task force, we’re getting a lot of real information that the general public doesn’t get… and yes this thing is very contagious, but using proper care is not a big deal. I think sometimes that the media plays it up. So sometimes it’s good not to watch the news. (#9, 73 y/o Caucasian, male living with spouse in a house)
Media lacking	I’m just saying, when it comes to the health of the older people, they will tell you OK stay at home, do this… but nobody tells you, OK should you get more vitamins, should you eat this, or don’t eat that… you don’t hear that. (#18,78 y/o Black female, lives alone in a house)
Media lacking	I find that the news is not really giving us good information…we are finding that the information is not very clear at all, even with government websites, certainly not in the newspapers. (#13, 73 y/o Jewish male, lives with spouse in a condominium)
codes	Subtheme 4: Living Arrangement Challenges
Living alone	Even if something small broke down like my smoke detector… it started beeping and I couldn’t get anybody to come in to take the stupid thing down and put in a new one because no one was coming into the house and that was true for anything really, anything that you needed done. It’s very much that feeling of helplessness. (#15, 65 y/o Ukrainian female, lives alone in a house)
Living alone	That’s the biggest thing for me is that living alone I have quite frankly started to name the spiders in my house. I don’t really have a bubble. So that’s the thing that I’m kind of missing. That everybody has made, created these little bubbles with their immediate family, ‘cause those are the people that they really need. They need to bubble with, and I totally get that, but when I don’t really have family in the city, it’s harder for me to have a bubble. And that’s the thing that I have found the hardest. (#3, 67 y/o Caucasian female, lives alone in a house)
Living with spouse	People in our condo who are widowers, and some of them have trouble coping with being alone. So even though we’re not together all the time in the condo, we’re in our separate rooms at times, you know that person is there, and you can discuss what you like about what they’re showing on [TV show] or things like that, or have a friendly competition of answering the Jeopardy questions. (#11, 74 y/o Jewish male, lives with spouse in a condominium)
Living in small space	We’re more cross with each other and I think that’s just inevitable. we’re together 24 h a day, most of the time, and that’s too much for anybody. (#7, 81 y/o Caucasian male, lives with same-sex spouse in an apartment)
Living in small spaces	If you catch the virus, you need to quarantine yourself. But because of the construction of our townhouse. If one of us catches it, well then the person has to quarantine. But what about the other two? They might be in danger of catching the virus. (#1, 75 y/o Filipino male, lives with spouse and children in a house)
Living in large spaces	We’re not in a one-bedroom apartment, we’re in this large house that we share. We’re in the fortunate group that can do that, if I want to hide from her, I can go hide—if she needs to hide from me she can. I can hide out in the backyard…(#12, 70 y/o Caucasian male, lives with spouse in a house)
Living in large spaces	We’ve got a big backyard, we’re back on a ravine, so we’ve got privacy back there, we can go sit back there. We tend to sit out front, decked out front as well because the neighbors walk by and you talk to people. And its big enough so that we can have someone on the deck and still be social distanced by 6 feet. Which we do. (#22, 65 y/o Caucasian female, lives with spouse in a house)
codes	Subtheme 5: The Challenge of Physical Distancing and Minimal Social Interactions
Missing social groups	Gone are the days sitting around eating a meal, going to the movies, all that stuff. I mean, normally we would like to get together, go for a walk, or a swim, and then go eat or have a coffee, and we would walk in the valley or somewhere nice. That’s gone. (#20, 67 y/o Caucasian female, lives with spouse in a condominium)
Missing social groups	Social distancing, I think, is probably the one that has affected me more because I cannot go out with my friends, we can’t gather together…it makes me a little sad that the things I used to do, go out and have a cup of tea or coffee with my friends, I cannot do that. Before COVID, maybe three or four of us will go for a coffee or we go for lunch or we do things together, and I enjoy that very much. (#18 78 y/o Black female, lives alone in a house)
Missing social groups	All my social interactions like my group that I’m involved with and volunteering at [institution] is all cancelled so that was kind of hard because you don’t have any social outlets that way. (#2, 74 y/o Caucasian female, widow lives alone in a condominium)
Missing social groups	I want to go to a restaurant and eat with other people. That’s my biggest thing now. Yeah I’m dying- and I want to get out of the city. I don’t have a car. So it’s more difficult for me to get out. And even if I get out, you know, where would I go because I don’t know, you know, just because Toronto is still been kind of a hot spot. And a lot of my friends have cottages. So they just disappear for the summer. And usually I get invited to the cottage… but now, they’re kind of isolating themselves at cottages. (#3, 67 y/o Caucasian female, lives alone in a house)
Missing places of worship	Before the pandemic, I was in the synagogue twice a day and, you know, communing with a group of people… but some of it has had to be completely cancelled and I will say I have adapted, but it’s still a big change in my circumstances. (#13, 73 y/o Jewish male, lives with spouse in a condominium)
Missing places of worship	I was an usher in my church but I have given it up. Ushering is something that I really like to do and chit chat with people at the back of the church, and yeah I think in a sense I have missed that. (#18, 78 y/o Black female, lives alone in a house)
Missing places of worship	I miss going to church, I’ll be honest. But they did open our parish and I chose not to be one of the ones that went back. I don’t think it’s time. I think it’s too soon. I can watch it online. It’s not as good, but it certainly helps. So that’s a big part of my life because I’m very strong Catholic. (#5, 74 y/o Caucasian female, lives with spouse in a house)
Missing family	We cannot see our grandchildren up close and personal because, well because they [the parents] don’t want us, they don’t want to risk our being with the kids… it’s been an emotional loss exacerbated by the COVID thing. (#7, 81 y/o Caucasian male, lives with same-sex spouse in an apartment)
Missing family	Psychologically we’re missing our contact with our family…I’ve got grandchildren, my son, his wife is strict over the whole rules and she hasn’t stepped foot in our house since March… I wouldn’t say it’s me being depressed or anything but it’s disappointing. (#13, 73 y/o Jewish male, living with spouse in a condominium)
Missing family	I can’t go and hug my own grandchildren… It really upsets me a lot that my grandchild doesn’t know us and he’ll never really know us the way my grand-daughters do because he doesn’t see us. (#14, 70 y/o, Irish Jewish female, lives with spouse in a condominium)
Need for human contact	Now, I’m a hugger. It’s painfully hard for me to not run up and throw myself all over someone. I can do the elbow thing now. I’m learning. This is a day-by-day thing I will not lie. I’m not ashamed to say this. I have gone in the bedroom after the bed is all made later in the day and that [body] pillow is there and I have actually laid down and I’ve hugged it and I’m not ashamed to say that. And there is a little bit of comfort and, you know what, you take what you can get. Sounds ridiculous, but whatever works. Everybody should have a pillow. (#5, 74 y/o Caucasian female, lives with spouse in house)
Need for human contact	I’ve never really considered myself like a really overly affectionate person but I’m just really, you know, I’ve been out now for a massage and I went and had my nails done yesterday and I had my hair done last week and it’s like just to have another person touch me has been fantastic! (#3, 67 y/o Caucasian female, lives alone in a house)
Need for human contact (emotional wellness)	Not being able to see my friends has made it harder for managing my post-traumatic stress… I have to be very careful about what I read and what I watch on movies or TV or whatever, that it doesn’t trigger me so being in COVID and not being able to see some of the people that I miss has made it a little harder. (#8, 71 y/o Caucasian female, lives with family in group housing)
Life event	Well, I live on my own now, which is a big change. So that’s an adjustment for me because cooking for one… and what I do is I put the radio on as soon as I wake up, and that helps because somebody is talking, you know? I’m so happy for my radio. [My husband] didn’t die of COVID-19 it was just his heart gave out, you know? In the hospital, I could not visit because of the COVID and so his decision was to have palliative care, which in a way was really good because he was with me, you know? … I make sure I have a distance which is kinda sad in a way, but that’s the way it is right, can’t be hugging anybody… I find that very hard especially after a loss you know. (#2, 74 y/o Caucasian female, lives alone in a condominium)
Life event	My wife was admitted to hospital and for 12 days I couldn’t visit her... I was worried and I worried that she was feeling lonely, but then the procedure she had had a positive effect on her health so I’m glad with the end result. (#17, 77 y/o Middle Eastern male, lives with spouse in condominium)
Life event	And during that week I was supposed to have a retirement party—three of them scheduled for the last week. And so when I went home, everything was cancelled. I didn’t get to say goodbye to my friends. Some of them probably didn’t even know I was retired. Everything just went poof and it was gone. (#1, 75 y/o Filipino male, lives with spouse and children in house)
Not missing it	In a way it doesn’t bother me as much because I like being on my own sometimes, and quiet, and I don’t need interaction constantly. I’m just happy to be with me. (#2, 74 y/o Caucasian female, widow lives alone in a condominium)
Not missing it	My social sphere is restricted dramatically…my life isn’t noticeably worse because I don’t get to spend a couple afternoons at the senior centre. Now for some people though, that’s their whole world, and they’re there every day all day long. (#12, 70 y/o Caucasian male, lives with spouse in a house)
codes	Subtheme 6: The Challenge of Health Management and Health Services
Maintaining health routine	I am a member of the [community center] and that was my outlet for exercise [for my scoliosis]. I went to three-to-five classes for aquafit and I swam approximately a mile a week—that’s gone. So I’ve been more than half a year without this wonderful exercise that made me feel a lot more energetic than I am now. (#7, 81 y/o Caucasian male, lives with same-sex spouse in an apartment)
Transient health concern	For a couple of weeks ago I was having some difficulty sleeping and I actually talked to my doctor about it and she had suggested that I try melatonin. … I think it was just, not stress so much as just feeling tired of the sameness of it all. You know, just worn down a little bit. (#3, 67 y/o Caucasian female, lives alone in a condominium)
Exacerbation of existing symptoms	You still get the intrusive thoughts [obsessive compulsive disorder] but you don’t have to act on them. But I noticed it has been exacerbated during this pandemic. (#20, 67 y/o Caucasian female, lives with spouse in a condominium)
Exacerbation of existing symptoms	The fact that I have some anxiety, certainly the anxiety intensified because I didn’t have the systems to be able to talk to. (#6, 71 y/o Caucasian female, lives alone in a house)
Health services	I used to go just over half a block away and go see my doctor and get appointments within the week and now you have to go through all these gatekeepers, you have to go through somebody on the phone that doesn’t know you, and then I talked to the nurse who does know me and then she talked to the doctor and, you know, so it’s levels and levels of bureaucracy to get to your doctor. (#10, 72 y/o Celtic Jewish female, lives with family in group housing)
Health services	One of thing that I find really frustrating is that I cannot see a doctor [for chronic pain condition]. I can have a telephone appointment, but what I needed was for somebody to examine me… and it leads to a lot of frustration for me and a lot of anxiety and it’s also hard to diagnose the problem. (#15, y/o Ukrainian female, lives alone in a house)
Health services	I have had access to my family doctor. I have had a phone call with him, then I have also had a phone call with a specialist because my family doctor has sent me to a specialist. It did go well but I don’t think it was sufficient. If I was sitting in his office, I’m pretty sure he would have asked me a few more questions. (#18, 78 y/o Black female, lives alone in a house)
Health services	I had great support from the palliative care team [that supported husband] and I must say, I’m okay, the support was good. I’m lucky that I’m quite healthy, so I didn’t have to go to too many appointments. And you know, most of my appointments are for my eye, and so one is cancelled because they were not very important, or delayed. So I was lucky to be in good health, so it didn’t affect me much that way. (#2, 74 y/o Caucasian female, widow lives alone in a condo)
Health services	I’ve had a Zoom doctor appointment and I’m concerned a little bit about how they’ve managed anything if they physically needed to see me. I think we’ve had some diminishment in medical attention, obviously they’ve been focusing on people who are much sicker. (#13, 73 y/o Jewish male, lives with spouse in a condominium)
codes	Subtheme 7: Use of Technology: A New Necessity
Need for new learning	I’ve had to bunk up on my technology, I’ve had to learn how to do zoom, I’ve had to learn how to photograph a check and not go the bank… So, we’re adapting to more of an online presence in our life, for everything—banking, talking to people, zooms. I’m more terrified of going out in public than I am of technology. (#22, 65 y/o Caucasian female, lives with spouse in a house)
Need for new learning	It was a learning experience. I don’t really one hundred percent like it as much. Like playing bridge online, or doing book club online, you know what I mean, it’s not as nice as being in a group setting that you have each other, you know. It’s okay… but I wouldn’t like it to have it all the time like that. I don’t want it to stay forever that way. In the meantime, I adjust. (#2, 74 y/o Caucasian female, widow lives alone in a condominium)
Need for new learning	FaceTime was a great support, again being my age and not being used to the social media and how to navigate that was a process as well, but those supports were there and definitely, definitely helpful. (#6, 71 y/o Caucasian female, lives alone in a house)
Need for assistance	Getting the internet set up was a little, but no, not too much stressful. Like, I had to call the company, like Rogers, several times and get them to help and come and set it up. And my nephew had to come and aid and get everything going with me. A little bit inconvenient but not stressful. (#18, 78 y/o Black female, lives alone in house)
Need for assistance	Some older adults don’t have the means to really communicate. For example, they don’t have computer skills, Zooming, and so on... Some need help, encourage them to be more active… help them find the means of communicating with others. There was one person but then she was adamant she didn’t want to use the Zoom. She thought it’s so complicated and she didn’t have a camera, she didn’t have a microphone. I really tried to push her and finally she is now happy that she is doing it, so there are people to help… These people don’t have the skill and some of them really not forthcoming they are not… because they think they are too old to learn these new techniques. (#17, 77 y/o Middle Eastern male, lives with spouse in a condominium)
Resistance	I have a very close friend and she is nagging me like crazy to do Zoom because she loves Zoom. I’m willing to try it just so that we can connect, but also because everything seems to be on Zoom. I should learn how to use it now, but I also should learn how to use it for what’s coming up because everything seems to be on Zoom. (#15, 65 y/o Ukrainian female, lives alone in a house)
Resistance	I’m not loving zoom, but it’s become kind of necessary I guess in all of this. (#3, 67 y/o Caucasian female, lives alone in a house)
Better than nothing	It was a learning experience. I don’t really one hundred percent like it as much. Like playing bridge online, or doing book club online, you know what I mean, it’s not as nice as being in a group setting that you have each other, you know. It’s okay… but I wouldn’t like it to have it all the time like that. I don’t want it to stay forever that way… In the meantime, I adjust. (#2, 74 y/o Caucasian female, widow lives alone in a condominium)
Better than nothing	I don’t get the same sense of encouragement on Zoom that I would on a face-to-face event. I mean it’s better than nothing but I’m not a big fan of this new technology, I’d rather have an actual dialogue with people. (#8, 71 y/o Caucasian female, lives with sister in co-op)
Unexpected benefits	It’s actually better in some ways because in book club, you get all these, it’s hard to hold focus “let’s talk about the book”. You got these two over here talking and those two over there talking and you’re like “excuse me, excuse me”. So, it’s actually easier when you’re all on a screen. (#22, 65 y/o Caucasian female, lives with spouse in house)
Unexpected benefits	The thing is we can’t see them [i.e., mom in nursing home]. We could see them through the window, which is fine. But now you can’t see them at all. So, this is what we do three times a week, I’m on Facebook with my mom, so thank God for that. (#4, 69 y/o Caucasian female, lives with spouse in a house)
Unexpected benefits	I mean to be honest, it’s kind of nice that all these speakers are coming to me, and I don’t have to drive all over the place, or get on the TTC to hear this speaker, or that speaker. (#20, 67 y/o Caucasian female, living with spouse in a condominium)
Unexpected benefits	We both have our own laptop, so we can reading, listening, musical arts performance is more than before. Free, hahahah! Specially opera, which I am really fanatic about, and he likes the sports, so we have own time to do that without much boredom or anything. (#19, 78 y/o Korean female, lives with spouse in a condominium)
**Theme 2: Coping with the COVID-19 Pandemic**	
codes	Subtheme 1: Behavioral strategies
Mask wearing and other precautions	Because I’m asthmatic, I was wearing the disposable masks, I really had trouble breathing. But I was determined to find a mask I could wear, and since then, we’ve found cloth masks that don’t bother my breathing so much. I tried to find something that would work instead of just saying ‘well I can’t wear a mask,’ because I feel very vulnerable with my autoimmune disease. I’m glad that I found that because I feel protected and I’m protecting other people from me too. (#8, 71 y/o Caucasian female, lives with family in group housing)
Mask wearing and other precautions	I believe the physical social distancing, cleaning hand, to clean more surface…I’m a chemistry major, so I really believe those directions. Very naïve way, but [that is] only [what] they can suggest now without the vaccine development. So, I trust all the direction. If I follow, I am more safe. (#19, 78 y/o Korean female, lives with spouse in a condominium)
Minimizing media	I was listening to that [CBC news] all the time, all the shows, you know. I would have it on all day. So that I’ve eliminated, and I think that I’m less anxious because of that. (#10, 72 y/o Celtic Jewish female, lives with family in group housing)
Minimizing media	I don’t watch the news now. I catch the headlines of an article on my iPad. I keep up on it enough that way, for me. I don’t need the details every single day. It’s too disturbing. (#20, 67 y/o Caucasian female, living with spouse in a condominium)
Minimizing media	It’s loaded. It’s too much. It’s overload. Too much news, too much information sometimes. That’s why I shut it off… (#2, 74 y/o Caucasian female widow, lives alone in a condominium)
Making a schedule	I try to make sure I have a schedule. I get up, I do a fitness routine, I get dressed, I have a zoom something in the morning or, you know I really try hard to make sure that I do have the structure. And I also make sure that I get out every day because I think, first, the fresh air…so I see lots of green, which is very important for me. (#21, 75 y/o Jewish female widow, lives alone in a condominium)
Making a schedule	Each day I would have a goal for myself and usually when I do that I’m home, I get up, let’s do this but let’s get out, but I found that I really had to force myself some days to do things that that were on my list, so I did them but I didn’t do them with a great deal of enthusiasm. (#6, 71 y/o Caucasian female, lives alone in a house)
Pivoting	It didn’t really impact my wellbeing at all, it was just a shift in what to do with my available free time. Instead of sitting around and moping, you know, I walked more, biked more, made it home more, watched the news. It’s just a shift in how one occupies oneself, productively, stimulating. (#16, 70 y/o Jewish male, lives with common-law partner in a condominium)
Pivoting	One activity I’m doing that I wasn’t doing before, I started to learn to bake. I’ve started with scones, which I never did before, and we’ve been watching the great Canadian and Great British baking show, and that kind of inspired me to, you know, sort of challenge for me, so in a positive way that’s affected that. (#10, 72 y/o Celtic Jewish female, lives with family in group housing)
Pivoting	One positive aspect is that I sort of did activities that I wanted to do before but didn’t have the time, like trying to improve my language and my voice. (#17, 77 y/o Middle Eastern male, lives with spouse in a condominium)
Keeping busy	I have more time for myself now—to do whatever I want to do. I have more things to do now. (#1, 75 y/o Filipino male, living with spouse and children in a house)
Keeping busy	All kinds of things are popping up in my head that now are filling time for me. I think the more you open your mind to what you could be doing, the pandemic takes a little back seat for that while. Gardening, I think, takes my mind off a lot of other things, cuts about the weed. I think that’s all healthy stuff…I think our busyness helps. (#5, 74 y/o Caucasian female, lives with spouse in a house)
Not Pivoting (Importance of being self-directed)	Well because it’s like you have no purpose, like you can’t go do the things you normally would do, you know, like you know, on a Monday I did this, I went here and I… all that stuff I can’t do. I can’t socialize with a group of friends and go play cards or do whatever because we can’t do it right now, you know, so anything you planned or look forward to, you really can’t do, you know. (#4, 69 y/o Caucasian female, lives with spouse in a house)
Physical activity	I still try to do things that I was doing before the pandemic, you know. Even though I might go for a walk and I know my knees are gonna be ached, I’ll still go out. (#11, 74 y/o Jewish male, living with spouse in a condominium)
Being outdoors/Physical activity	We do most of our walking in the morning, if it’s a nice day and by that I mean 24–25 degrees, we’ll walk in the afternoon sometimes, and I’ll be honest our deck overlooks a walking path and we see a lot of seniors walking. They walk and they walk and it’s a way to break up the monotony of being indoors. (#9, 73 y/o Caucasian male, lives with spouse in a house).
Being outdoors	The garden has been a tremendous comfort to counteract all that other negative stuff. (#7, 81 y/o Caucasian male, lives with same-sex spouse in an apartment)
codes	Subtheme 2: Emotion-focused strategies
Hope and Optimism	You have to kind of believe that it’s going to be OK because I think if you’re a negative person and you just go to bed at night and go you know it’s all for naught. Well, that’s tough. I think that’s a hard way to live, and I think that’s going to make people ill. (#5, 74 y/o Caucasian female lives with spouse in a house)
Hope and Optimism	I make the best of whatever I can do, always have. I’ve always, shall we say, risen to the occasion, no matter what it is, whether it’s some kind of an emergency or whether it’s this. (#14, 70 y/o Irish Jewish female, lives with spouse in a condominium)
Hope and Optimism	It’s just part of my personality, I don’t give into fear. You know, even in earlier times in my life when I had back issues, for example, and you know like I was bedridden for a while, but I just wouldn’t give into it, you know. I just felt like, ‘OK, well let’s figure out a way to recover from this and get on with life’, right. I just don’t give into that kind of thing or if I’ve had you know bad things happen, I won’t live my life in fear. (#3, 67 y/o Caucasian female, lives alone in a house)
Gratitude and small pleasures	I think the more you open your mind to what you could be doing, the pandemic takes a little back seat for that while… We’re both ice cream people, so we just go get an ice cream and sit in the car and eat it and come home, you know. I think you have to pick your pleasures. They’re little, but they’re not as little as they used to be. Now getting an ice cream cone is a big deal. (#5, 74 y/o Caucasian female, lives with spouse in a house)
Gratitude and small pleasures	I’m just so grateful that I did have a home over my head, I had lots of food, financially I wasn’t stressed, so I’m very happy about that. (#6, 71 y/o Caucasian female, lives alone in a house)
Acceptance	I’m not apprehensive so much as accepting that it [subsequent wave] will happen and to just work my way through it when it does. (#14, 70 y/o Irish Jewish female, lives with spouse in a condominium)
Acceptance	If I die, I die. I mean at this age it’s kind of moot as to whether or not and it’s gonna happen and I’m sort of resigned to that fact. I’ve had cancer, I’m now not getting enough exercise which I think every day lessens my chance of a longer life, but it’s all sort of, you know, benign acceptance rather than intense worrying. I’ve had a wonderful life and I don’t really mind that I’m closer to death than most people are. (#7, 81 y/o Caucasian male, living with same-sex spouse in an apartment)
Perspective taking	I tend to be a very positive person and because I’m also used to living alone, it’s not as big a change for me… I’ve had enough tragedies in my life that you see, different people adapt to things differently. Sometimes when you’re faced with a difficult situation you grow; other times you just fall apart. Because I’ve been through a lot of things I try to reach out to people who might need a helping hand because, again I really am very lucky. I have wonderful family and friends, and it really makes a big difference. (#21, 75 y/o Jewish female, widow lives alone in a condominium)
Perspective taking	Transplant patients have learned to look after themselves... and so I don’t know! I guess I’m so grateful to be alive that I’m going to enjoy every minute of it. The gratitude that I have today is much greater than I ever had before…possibly the experience that I’ve gone through, I’ve had a second chance at life, like I should be dead, right?…and so you have a choice, you can either bounce off the walls or you can enjoy what you got! Yeah we’re living through a tough time right now, you have a choice to enjoy it or not, I choose to enjoy it. (#9, 73 y/o Caucasian male, lives with spouse in a house)
Perspective taking	Our grandparents were refugees, and their parents had to deal with the pilgrims, and they had to come here with a suitcase. Also, in the Irish side, our grandfather came here and his father died. I think our family has ingrained in our DNA ‘just get on with it, and do what you have to do to keep on going,’ and I think that really helps in the pandemic time. (#10, 72 y/o Celtic Jewish female, lives with family in group housing)
Perspective taking	We came here at a vibrant age. Immigrant here. So we face difficulty in mid-20s. So maybe not like pandemic COVID-19 hardship, we had already hardship without the language, plus some level of racism and then during that time I was pregnant. (#19, 78 y/o Korean female, lives with spouse in a condominium)
Perspective taking	We haven’t had all the negatives that other people have experienced, we don’t have little kids to worry about we don’t have to homeschool, we don’t have to worry about if we’re going to keep our jobs. My age is such that I don’t have the physical infirmities that will cascade upon me when I hit 85 or 90. I’m in that lovely part of the seniors spectrum where I’ve got all the advantages and none of the disadvantages. (#12, 70 y/o Caucasian male, living with spouse in a house)
Meaning-making	There is a COVID-19 pandemic versus a racism pandemic. But this COVID-19 pandemic gave me more voices about the decision… I have more facts to [relay] to other people in my community who are kind of ignorant with subtle racism underneath. So, we need more education about it. Its all relatable. Its nothing separate issues. (#19, 78 y/o Korean female, lives with spouse in a condominium)
Silver lining	It’s given me a different perspective on things, on what things are truly important. Small things that you take for granted that are important to your well being, a different perspective on that, I’m so grateful for those things. (#6, 71 y/o Caucasian female, lives alone in a house)
Silver lining	People have stepped up to the plate to try to adjust to new circumstances and they’ve done remarkably well in my opinion…this challenge [the pandemic] has really made us see what wonderful people that we have in our community. (#13, 73 y/o Jewish male, lives with spouse in a condominium)
codes	Subtheme 3: Social Support
Being connected	So, part of the ease of going through the pandemic is the way we’re all connected. Fifty years ago, how would you have done? (#12, 70 y/o Caucasian male, lives with spouse in a house)
Being connected	It’s just maybe three women sharing three different feelings, and most of those conversations end up in all three or all four of us being a whole lot better because, you know, you’re not the only one feeling it. You know, you’re not the only one that doesn’t have the answers, and that gets scared sometimes. But I find my outlet with my friends always makes me stronger and I think we all we all hang up and go, ‘Yeah I can do this. I feel better’. (#5, 74 y/o Caucasian female, lives with spouse in house)
Supported by spouse	He’s definitely the one that will talk me down from something. So, if I say, you know, “What if?” He’ll just say, “Don’t what if it.” Just, you know, we’ll cross that bridge. He’s a lot more sensible and not up and down like I am. (#5, 74 y/o Caucasian female, lives with spouse in house)
Need for more support	I am always wishing for more support and I, so that’s you know, that’s an ongoing theme for me…my hopes and my wishes for you know social interaction and for people, you know coming to take care of me is unrealistic, and it was unrealistic before the pandemic. It’s great that my brother would come every Saturday but I also wanted him to phone me twice a week. Or even my friend in [city], I would hope that she would call me more than once a week … I think people have their own lives, I don’t think that’s completely realistic but that’s what I would want. (#15, 65 y/o Ukrainian female, lives alone in a house)
Supporting others	Father Vincent [pseudonym] sends me little jobs… my capabilities on the Internet are limited and he knows that so he’ll give me a list of calls to parishioners and just say ‘I’m calling on behalf of Father Vincent who just wanted to know how are you doing? Do you need anything?’ You know, so that’s a blessing. ‘Cause if you call 5 people in a day and you help one person who’s alone and say to them, ‘Listen, I’m calling on behalf of the perish, but you can have my phone number, and if you just wanted to chat you know I’m home, I’m stuck here too’. I’ve had a few calls, and I have to tell you, if you ever think that that’s not a blessing, you’re crazy, because that will take your mind off anything that you’re feeling. A couple of times, I talked to a senior, and I don’t know if I made them feel better, but I felt better. It was like I was talking to me. (#5, 74 y/o Caucasian female, lives with spouse in a house)
Supporting others	I would just take a little chair, sit outside her window when they open up the top so we could hear each other talk you know, stuff like that. … like not being able to go in and explain to her, that’s been my biggest… breaks my heart! because she doesn’t understand…and then she’ll say like there’s nobody here, nobody comes, and stuff like that, ‘cause they’re all confined to their rooms so they don’t leave the room at all… and she hasn’t left it in three or four months so that’s hard. (#4, 69 y/o Caucasian female, lives with spouse in a house)

## Data Availability

To protect the identity of participants, additional qualitative data can be made upon request.

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
