# Peer review of "Stress and Adjustment during the COVID-19 Pandemic: A Qualitative Study on the Lived Experience of Canadian Older Adults"

_ijerph, 2021, doi:10.3390/ijerph182412922_

Round 1

Reviewer 1 Report

Overall, this is a well written and structured paper which reports on key themes identified in a sample of Canadian, community dwelling, older adults, accounts of the early months of the COVID 19 pandemic. I feel the paper makes a helpful contribution and complements the growing qualitative literature on the topic of older adult’s experiences of the pandemic in high-income countries. Please consider the suggestions below which I believe would strengthen the manuscript:    

  • You need to explain the Canadian context of the study more clearly, to tailor the paper to the international audience of the journal. It is not until the Discussion that it is explained the sample were from Canada and this important detail needs to be included a lot sooner. I would suggest the Canadian context needs to be included in the title and somewhere in the abstract. In the introduction of the paper the authors need to explain the phases and dates of restrictions implemented in Canada and also other aspects of the pandemic response that they refer to e.g. bubbles. I would suggest it needs to be clearer what the regulations were in Canada at the time of data collection, or during the period that the participants are reflecting on. Also, please consider terms that are used in the paper e.g. Condo, which may not be familiar to readers outside of Canada.

  • Materials and methods

It is noted that the sample was recruited through online adverts, I would like more detail about these adverts (where online was the study advertised e.g. was it social media?) and please briefly note how the study was described in these adverts. Also, something to consider at some point in the paper (possibly in the limitations) is that one of your conclusions relates to technology/ internet use. Does the fact that the sample was recruited through the internet suggest this sample already had a level of confidence with technology?

You note that you reach saturation in the development of themes, however, from the description it appears that all the analysis was undertaken after data collection, could the stages of data collection and analysis be explained slightly more clearly. I would argue that the use of the term data saturation is now contested, and data sufficiency may be more appropriate. If you do choose to continue to use the term saturation, please could a reference be included so the reader can understand how you are using this term.  

Please could you include the semi-structured topic guide as an appendix so it is clearer how the questions asked relate to the themes that were identified.

Please take out the term ‘themes emerged’ from the abstract and lines 117 (and anywhere else in the manuscript it is used) and replace with ‘themes were identified’ The use of ‘themes emerged’ is not considered good practice in qualitative reporting as it suggests no researcher interpretation was involved in the development of the themes.

While you explain the sample in the table, I would like you to narratively describe the sample i.e. this sample are predominantly partnered, and only a small number were living alone. It was helpful to see your consideration of socio-economic status at different points in the manuscript (health inequalities related to socio-economic status / social class have been profound during the pandemic), what measure of socio-economic status did you use?

  1. I do not think it is necessary (or usual practice) to include all the data / quotes in a supplementary appendix. Your analysis approach appears rigorous and is well explained, and you include a lot of quotations in Table 2 in the manuscript.  Thus, no one should query the data that has led to the findings. I suggest that noting in the ‘Data Availability Statement’ that the full data set can be made available upon request, is sufficient. I also think by including that amount of data for interviews with the participant identifiers in the appendix you may start to compromise participant anonymity. 
  2. Overall, the Discussion seems well structured and the conclusion is interesting. I think it would be helpful at some point to compare the findings more explicitly to other qualitative studies with older adults in other countries during the pandemic. There are some in this journal and a google scholar search indicates there is work in other countries including the UK. Your oldest participant was 81, it might be worth noting that the sample did not include the oldest older adults and consider how their experiences may have differed.  

Please check the manuscript for typos.  

Lines 36-37 – what time period were these 26,000 calls made in?

Author Response

We sincerely thank the reviewers for their time and consideration in reviewing our manuscript. The feedback provided was insightful and helped to improve the quality of the revised manuscript. Please find our itemized response to each of the review comments below.

Of note, we have indicated the location of various revisions with reference to the “simple markup view” (i.e., clean view), rather than the “all markup view”. We realize that these terms may differ depending on the program used by the viewer.

REVIEW 1:

Overall, this is a well written and structured paper which reports on key themes identified in a sample of Canadian, community dwelling, older adults, accounts of the early months of the COVID 19 pandemic. I feel the paper makes a helpful contribution and complements the growing qualitative literature on the topic of older adult’s experiences of the pandemic in high-income countries. Please consider the suggestions below which I believe would strengthen the manuscript:    

Response: We thank the reviewer for their time in reviewing the manuscript and for providing critical feedback to help strengthen the paper.

You need to explain the Canadian context of the study more clearly, to tailor the paper to the international audience of the journal. It is not until the Discussion that it is explained the sample were from Canada and this important detail needs to be included a lot sooner. I would suggest the Canadian context needs to be included in the title and somewhere in the abstract.

Response: We thank the reviewer for this comment. As suggested, we have specified Canada in the title of the manuscript (Line 3 on the Simple Markup View) and in the abstract (Line 15 of the Simple Markup View).

In the introduction of the paper the authors need to explain the phases and dates of restrictions implemented in Canada and also other aspects of the pandemic response that they refer to e.g. bubbles. I would suggest it needs to be clearer what the regulations were in Canada at the time of data collection, or during the period that the participants are reflecting on. Also, please consider terms that are used in the paper e.g. Condo, which may not be familiar to readers outside of Canada.

Response. We thank the reviewer for this suggestion. The revised introduction includes a brief description of COVID-19 regulations that were in place during the study timeline (Lines 27-41 of the Simple Markup View). We also clarify that participants, recruited between May and October of 2020, were asked to reflect on their experience since the beginning of the pandemic (Lines 89-90 of the Simple Markup View).

 We also appreciate the note on using certain terms that may not be familiar to an international audience, such as condo and social bubbles.

Materials and methods: It is noted that the sample was recruited through online adverts, I would like more detail about these adverts (where online was the study advertised e.g. was it social media?) and please briefly note how the study was described in these adverts.

Response: As requested, we have provided more detail on study recruitment and the study advertisement [Lines 80-86 of the Simple Markup View].  We also note that snowball sampling was included, as some participants shared the advertisement with their peers.

Also, something to consider at some point in the paper (possibly in the limitations) is that one of your conclusions relates to technology/ internet use. Does the fact that the sample was recruited through the internet suggest this sample already had a level of confidence with technology?

Response: We thank the reviewer for this comment. We would like to note that two of our participants completed the interview via telephone due to accessibility and comfort with Zoom. We have included this note in the revised manuscript [Lines 95-96 of the Simple Markup View]

As suggested in the transcripts, various experiences were shared with respect to technology use and comfort with using technology. Some participants reflected on their friends who were resistant to learning Zoom and other communication platforms, acknowledging that this resistance may contribute to a sense of loneliness. Some participants noted that they did not necessarily feel comfortable using online platforms, but saw it as a necessity. The discussion further expands on the health benefits of internet use among older adults [please refer to Lines 569-571 of the Simple Markup View]. However, we have integrated this comment with the one noted below concerning age limit within our revised limitations section (see below)

You note that you reach saturation in the development of themes, however, from the description it appears that all the analysis was undertaken after data collection, could the stages of data collection and analysis be explained slightly more clearly. I would argue that the use of the term data saturation is now contested, and data sufficiency may be more appropriate. If you do choose to continue to use the term saturation, please could a reference be included so the reader can understand how you are using this term.  

Response: We thank the reviewer for this comment. Data sufficiency was determined by the researchers after the 22nd interview. By the 22nd interview, the researchers agreed that no new information was being shared, with a pronounced replication in story themes. We then confirmed data sufficiency during the initial phase of coding. The statement of data saturation has been revised, using more appropriate terminology [Lines 96-99 of the Simple Markup View]

Please could you include the semi-structured topic guide as an appendix so it is clearer how the questions asked relate to the themes that were identified.

Response: The interview questions are included in the revised manuscript [Lines 111-119 of Simple Markup View]

Please take out the term ‘themes emerged’ from the abstract and lines 117 (and anywhere else in the manuscript it is used) and replace with ‘themes were identified’ The use of ‘themes emerged’ is not considered good practice in qualitative reporting as it suggests no researcher interpretation was involved in the development of the themes.

Response: We thank the reviewer for this comment. The term “emerged” has been removed and replaced with more appropriate terms throughout the manuscript.

While you explain the sample in the table, I would like you to narratively describe the sample i.e. this sample are predominantly partnered, and only a small number were living alone.

Response: The section on participant characteristics has been revised and expanded.  [Lines 132-159 of the Simple Markup View]

It was helpful to see your consideration of socio-economic status at different points in the manuscript (health inequalities related to socio-economic status / social class have been profound during the pandemic), what measure of socio-economic status did you use?

Response: A self-report perceived socio-economic status question was used. Participants were asked to report their perceived status according to 5 categories: low, low to medium, medium, medium to high, high. This description is included in the revised manuscript [Lines 102-103 of the Simple Markup View].

I do not think it is necessary (or usual practice) to include all the data / quotes in a supplementary appendix. Your analysis approach appears rigorous and is well explained, and you include a lot of quotations in Table 2 in the manuscript.  Thus, no one should query the data that has led to the findings. I suggest that noting in the ‘Data Availability Statement’ that the full data set can be made available upon request, is sufficient. I also think by including that amount of data for interviews with the participant identifiers in the appendix you may start to compromise participant anonymity. 

Response: We appreciate the reviewer’s comment and concern. We have removed the supplement table and have included the statement “To protect the identity of participants, additional qualitative data can be made upon request” under the Data Availability Statement.  Of note, Table 2 was revised to ensure that all codes/themes were fully represented and that there was adequate representation by sex, ethnicity and living arrangement.

Overall, the Discussion seems well structured and the conclusion is interesting. I think it would be helpful at some point to compare the findings more explicitly to other qualitative studies with older adults in other countries during the pandemic. There are some in this journal and a google scholar search indicates there is work in other countries including the UK.

 Response: We thank the reviewer for this suggestion. The revised discussion now refers to other relevant qualitive studies. The studies that we felt were most relevant include:

  • Swiss sample: [20] Falvo, I., Zufferey, M.C., Albenese, E. & Fadda, M. (2021). Lived experiences of older adults during the first COVID-19 lockdown: A qualitative study. PLOS ONE 16(6): e0252101.
  • UK sample: [25] McKinlay, A.R., Fancourt, D., & Burton, A. (2021). A qualitative study about the mental health and wellbeing of older adults in the UK during the COVID-19 pandemic. BMC Geriatrics, 21, 439.
  • United States: [30] Hamm, M.A., Brown, P.J., Karp, J.F., Lenard, E., Cameron, F., Dawdani, A., Lavretsky, H., Miller, J.P., Mulsant, B.H., Pham, V.T., Reynolds, C.F., Roose, S.P., & Lenze, E.J. (2021). Experiences of Americal older adults with pre-existing depression during the beginnings of the COVID-19 pandemic: A multicity, mixed-methods study. American Journal of Geriatric Psychiatry, 28(9), 924-932.
  • French sample: [34] Goethals, L., Barth, N., Guyot, J.,Hupin, D., Celarier, T., & Bongue, B. (2020). Impact of home quarantine on physical activity among older adults living at home during the COVID-19 pandemic: Qualitative interview study. JMIR Aging, 3(1), e19007.
  • United States: [36] Fuller, H.R. & Huseth-Zosel, A. (2021). Lessons in Resilience: Initial Coping Among Older Adults During the COVID-19 Pandemic. The Gerontologist, 61(1), 114-125. https://doi.org/10.1093/geront/gnaa170
  • Italian sample: [40] Cipolletta, S., & Gris, F. (2021). Older people’s lived perspectives of social isolation during the first wave of the COVID-19 pandemic in Italy. International Journal of Environmental Research and Public Health, 18, 11832.

Your oldest participant was 81, it might be worth noting that the sample did not include the oldest older adults and consider how their experiences may have differed.  

Response: We thank the reviewer for this observation and have included this as a limitation of the study. [Lines 596-600 of Simple Markup View]

Please check the manuscript for typos.  

Response: We have scanned and revised the document for typos.

Lines 36-37 – what time period were these 26,000 calls made in?

Response: The start date of March 9th 2020 is now included in the sentence [Lines 45-51 of Simple Markup View]

Reviewer 2 Report

This is a good paper and it makes a useful contribution to the body of knowledge.  It is important to catalogue the experience of older people in the early phase of the pandemic.  This qualitative approach does so well, mindful of limitations which are described.

While I believe the paper to be of very good standard, I wish to make some observations which may assist the authors in improving their manuscript.  These points are for consideration - I respect the editor and authors' decision as to whether they should be acted on.

1). In the excerpts, I noted that the interview dates were included with some and not with others.  It may be better to do this consistently for all interview or perhaps not at all.  Interview date may be unnecessary.

2). There is a good level of granularity in outlining the demographic detail of the participants.  Although very small, is there a risk of some participants being potentially identifiable?  For example - "70 y/o, Irish Jewish female,
lives with spouse in condo".  One might think given such good description the person might be identifiable to those who know her?  This is something to consider.

3). Sub-themes 3.2.1.2 Financial threat seems a little short and out of place compared to other larger sub-themes.

4). The excerpts are presented in table format and as a cohesive whole.  I feel this works well and I support it.  However, I am conscious that some readers might prefer or expect excerpts to be included in the text.  I do not recommend a change rather wish to draw attention to a different perspective.

5). The data analysis is appropriate.  Consideration could be given to the use of a Table or Figure displaying the number of codes and how these contributed to sub-themes.

6). While the discussion is good, I recommend greater use of literature to strengthen the degree of synthesis and to emphasise the important points being made.

Overall, I enjoyed this manuscript.  I feel it is a useful and important piece of research which is well written.

Thank you for the opportunity to review.

Author Response

REVIEWER 2:

This is a good paper and it makes a useful contribution to the body of knowledge.  It is important to catalogue the experience of older people in the early phase of the pandemic.  This qualitative approach does so well, mindful of limitations which are described.

While I believe the paper to be of very good standard, I wish to make some observations which may assist the authors in improving their manuscript.  These points are for consideration - I respect the editor and authors' decision as to whether they should be acted on.

Response: We thank the reviewer for their time in reviewing the manuscript and for their positive appraisal of the study.

1). In the excerpts, I noted that the interview dates were included with some and not with others.  It may be better to do this consistently for all interview or perhaps not at all.  Interview date may be unnecessary.

Response: We thank the reviewer for noting this inconsistency. In response to Reviewer 1, we have removed the Supplement Table which is where the interview dates were noted.

2). There is a good level of granularity in outlining the demographic detail of the participants.  Although very small, is there a risk of some participants being potentially identifiable?  For example - "70 y/o, Irish Jewish female, lives with spouse in condo".  One might think given such good description the person might be identifiable to those who know her?  This is something to consider.

Response: We appreciate this comment and the concern for participant confidentiality. However, we felt it important to include these characteristics to help contextualize the quotes. This was how the participant identified themselves and therefore we wanted to stay true to their identity. It is not uncommon for individuals to identify with their ancestral origin (e.g., Ireland) and their ethno-cultural identity (i.e., Jewish). For example, there are many individuals that identify as Polish Jewish, or Russian Jewish. This sample also included one woman who identified as Celtic Jewish. These identities are different from persons who identify as Israeli Jewish.   

3). Sub-themes 3.2.1.2 Financial threat seems a little short and out of place compared to other larger sub-themes.

Response: We thank the reviewer for this comment. We acknowledge that financial threat was not one of the richest themes generated. This is largely due to the fact that there few participants in the sample who were not retired and still working. Importantly, among those still working, financial threat was the largest concern experienced during the first 6 months of the pandemic. Accordingly, we felt that it was important to include financial threat as one of the subthemes. This section (Lines 203-212 on the Simple Markup View) and the brief discussion on financial threat (please see Lines 465-477 on Simple Markup View) are slightly expanded on in the revised manuscript.

4). The excerpts are presented in table format and as a cohesive whole.  I feel this works well and I support it.  However, I am conscious that some readers might prefer or expect excerpts to be included in the text.  I do not recommend a change rather wish to draw attention to a different perspective.

Response: We appreciate this comment and agree that there are different preferences among scholars. Our initial draft of this manuscript included quotes within the text; however, this resulted in a significantly longer manuscript. In considering length and to not overburden the reader, we chose to incorporate a table format instead.

5). The data analysis is appropriate.  Consideration could be given to the use of a Table or Figure displaying the number of codes and how these contributed to sub-themes.

Response: We would like to bring the reviewer’s attention to Table 2 which includes the study codes that were relevant to the themes/subthemes and excerpts.

6). While the discussion is good, I recommend greater use of literature to strengthen the degree of synthesis and to emphasise the important points being made.

Response: We thank the reviewer for this comment. The discussion has been revised, expanding on some of the discussion points that were made. As you will see, the revised discussion makes reference to other existing qualitative studies that were conducted in other countries during the initial wave of the pandemic.

Overall, I enjoyed this manuscript.  I feel it is a useful and important piece of research which is well written. Thank you for the opportunity to review.

Response: We thank the reviewer for this very encouraging note.

Reviewer 3 Report

the study is pretty good in its qualitative part including theme analysis. however, the quantitive part can be better even with few basic analyses of correlation between different questionnaire. moreover, if the authors can link the two part and suggest any findings showing statistical between group difference within various themes from the interview, that can elevate the research level.  

Author Response

REVIEWER 3

the study is pretty good in its qualitative part including theme analysis. however, the quantitive part can be better even with few basic analyses of correlation between different questionnaire. moreover, if the authors can link the two part and suggest any findings showing statistical between group difference within various themes from the interview, that can elevate the research level.  

Response: We appreciate the reviewer’s comment; however, we do not agree that the inclusion of correlations would contribute to the scope of the study. First, the sample (N=22) is too small to conduct meaningful statistical analyses. Furthermore, this qualitative study did not have testable hypotheses.